# Learning Task Specifications from Demonstrations

**Marcell Vazquez-Chanlatte**[1]**, Susmit Jha**[2]**, Ashish Tiwari**[2]**, Mark K. Ho**[1]**, Sanjit A. Seshia**[1]
[1] University of California, Berkeley    [2] SRI International, Menlo Park
{marcell.vc, sseshia, mark_ho}@eecs.berkeley.edu    {susmit.jha, tiwari}@sri.com

## Abstract

Real-world applications often naturally decompose into several sub-tasks. In many settings (e.g., robotics) demonstrations provide a natural way to specify the sub-tasks. However, most methods for learning from demonstrations either do not provide guarantees that the artifacts learned for the sub-tasks can be safely recombined or limit the types of composition available. Motivated by this deficit, we consider the problem of inferring Boolean non-Markovian rewards (also known as logical trace properties or *specifications*) from demonstrations provided by an agent operating in an uncertain, stochastic environment. Crucially, specifications admit well-defined composition rules that are typically easy to interpret. In this paper, we formulate the specification inference task as a maximum a posteriori (MAP) probability inference problem, apply the principle of maximum entropy to derive an analytic demonstration likelihood model and give an efficient approach to search for the most likely specification in a large candidate pool of specifications. In our experiments, we demonstrate how learning specifications can help avoid common problems that often arise due to ad-hoc reward composition.

## 1   Introduction

In many settings (e.g., robotics) demonstrations provide a natural way to specify a task. For example, an agent (e.g., human expert) gives one or more demonstrations of the task from which we seek to automatically synthesize a policy for the robot to execute. Typically, one models the demonstrator as episodically operating within a dynamical system whose transition relation only depends on the current state and action (called the Markov condition). However, even if the dynamics are Markovian, many problems are naturally modeled in non-Markovian terms (see Ex 1).

**Example 1.** Consider the task illustrated in Figure 1. In this task, the agent moves in a discrete gridworld and can take actions to move in the cardinal directions (north, south, east, west). Further, the agent can sense abstract features of the domain represented as colors. The task is to reach any of the yellow (recharge) tiles without touching a red tile (lava) – we refer to this sub-task as YR. Additionally, if a blue tile (water) is stepped on, the agent must step on a brown tile (drying tile) before going to a yellow tile – we refer to this sub-task as BBY. The last constraint requires recall of two state bits of history (and is thus not Markovian): one bit for whether the robot is wet and another bit encoding if the robot recharged while wet.

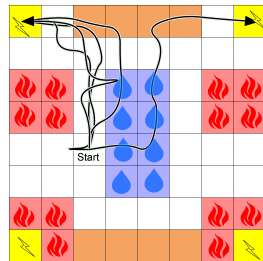

Figure 1

Further, like Ex 1, many tasks are naturally decomposed into several sub-tasks. This work aims to address the question of how to systematically and separately learn non-Markovian sub-tasks such that they can be readily and safely recomposed into the larger meta-task.

Here, we argue that *non-Markovian Boolean specifications* provide a powerful, flexible, and easily transferable formalism for task representations when learning from demonstrations. This stands in contrast to the quantitative scalar reward functions commonly associated with Markov Decision

Processes. Focusing on Boolean specifications has certain benefits: (1) The ability to naturally express tasks with *temporal dependencies*; (2) the ability to take advantage of the *compositionality* present in many problems, and (3) use of *formal methods* for planning and verification [29].

Although standard quantitative scalar reward functions could be used to learn this task from demonstrations, three issues arise. First, consider the problem of *temporal specifications*: reward functions are typically Markovian, so requirements like those in Ex 1 cannot be directly expressed in the task representation. One could explicitly encode time into a state and reduce the problem to learning a Markovian reward on new time-dependent dynamics; however, in general, such a reduction suffers from an exponential blow up in the state size (commonly known as the *curse of history* [24]). When inferring tasks from demonstrations, where different hypotheses may have different historical dependencies, naïvely encoding the entire history quickly becomes intractable.

A second limitation relates to the *compositionality* of task representations. As suggested, Ex 1 naturally decomposes into two sub-tasks. Ideally, we would want an algorithm that could learn each sub-task and combine them into the complete task, rather than only be able to learn single monolithic tasks. However, for many classes of quantitative rewards, "combining" rewards remains an ad-hoc procedure. The situation is further exacerbated by humans being notoriously bad at anticipating or identifying when quantitative rewards will lead to unintended consequences [11], which poses a serious problem for AI safety [1] and has led to investigations into reward repair [9]. For instance, we could take a linear combination of rewards for each of the subtasks in Ex 1, but depending on the relative scales of the rewards, and temporal discount rate, wildly different behaviors would result.

In fact, the third limitation - brittleness due to simple changes in the environment - illustrates that often, the correctness of the agent can change due to a simple change in the environment. Namely, imagine for a moment we remove the water and drying tiles from Fig 1 and attempt to learn a reward that encodes the "recharge while avoid lava" task in Ex 1. Fig 2a illustrates the reward resulting from performing Maximum Entropy Inverse Reinforcement Learning [35] with the demonstrations shown in Fig 1 and the binary features: red (lava tile), yellow (recharge tile), and "is wet". As is easy to verify, a reward optimizing agent, $\sum_{i=0}^{\infty} \gamma^i r_i(s)$, with a discount factor of $\gamma = 0.69$ would generate the trajectory shown in Fig 2a which avoids lava and eventually recharges.

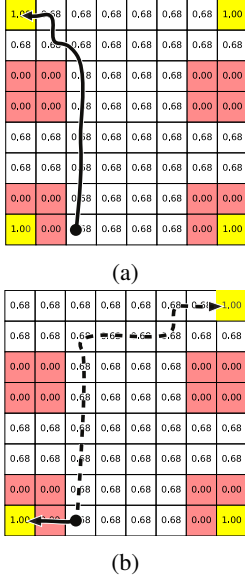
(a)

(b)

Figure 2: Illustration of a bug in the learnt quantitative Markovian reward resulting from slight changes in the environment.

Unfortunately, using the *same* reward and discount factor on a nearly identical world can result in the agent entering the lava. For example, Fig 2b illustrates the learned reward being applied to a change in the gridworld where the top left charging tile has been removed. An acceptable trajectory is indicated via a dashed arrow. Observe that now the discounted sum of rewards is maximized on the solid arrow's path, resulting in the agent entering the lava! While it is possible to find new discount factors to avoid this behavior, such a supervised process would go against the spirit of automatically learning the task.

Finally, we briefly remark that while non-Markovian Boolean rewards cannot encode all possible rewards, e.g., "run as fast as possible", often times such objectives can be reframed as policies for a Boolean task. For example, consider modeling a race. If at each time step there is a non-zero probability of entering a losing state, the agent will run forward as fast as possible even for the Boolean task "win the race".

Thus, quantitative Markovian rewards are limited as a task representation when learning tasks containing *temporal specifications* or *compositionality* from demonstrations. Moreover, the need to fine tune learned tasks with such properties seemingly undercuts the original purpose of learning task representations that are generalizable and invariant to irrelevant aspects of a task [21].

**Related Work:** Our work is intimately related to Maximum Entropy Inverse Reinforcement Learning. In Inverse Reinforcement Learning (IRL) [23] the demonstrator, operating in a stochastic environment, is assumed to attempt to (approximately) optimize some unknown reward function over the trajectories. In particular, one traditionally assumes a trajectory's reward is the sum of state rewards of the

trajectory. This formalism offers a succinct mechanism to encode and generalize the goals of the demonstrator to new and unseen environments.

In the IRL framework, the problem of learning from demonstrations can then be cast as a Bayesian inference problem [26] to predict the most probable reward function. To make this inference procedure well-defined and robust to demonstration/modeling noise, Maximum Entropy IRL [35] appeals to the principle of maximum entropy [12]. This results in a likelihood over the demonstrations which is no more committed to any particular behavior than what is required for matching the empirically observed reward expectation. While this approach was initially limited to learning a linear combination of feature vectors, IRL has been successfully adapted to arbitrary function approximators such as Gaussian processes [19] and neural networks [8]. As stated in the introduction, while powerful, traditional IRL provides no principled mechanism for composing the resulting rewards.

To address this deficit, composition using soft optimality has recently received a fair amount of attention; however, the compositions are limited to either strict disjunction (do X *or* Y) [30] [31] or conjunction (do X *and* Y) [10]. Further, because soft optimality only bounds the deviation from simultaneously optimizing both rewards, optimizing the composition does not preclude violating safety constraints embedded in the rewards (e.g., do not enter the lava).

The closest work to ours is recent work on inferring Linear Temporal Logic (LTL) by finding the specification that minimizes the expected number of violations by an optimal agent compared to the expected number of violations by an agent applying actions uniformly at random [16]. The computation of the optimal agent's expected violations is done via dynamic programming on the explicit product of the deterministic Rabin automaton [7] of the specification and the state dynamics. A fundamental drawback to this procedure is that due to the curse of history, it incurs a heavy run-time cost, even on simple two state and two action Markov Decision Processes. We also note that the literature on learning logical specifications from examples (e.g., [15, 33, 20]), does not handle noise in examples while our approach does. Finally, once a specification has been identified, one can leverage the rich literature on planning using temporal logic to synthesize a policy [17, 28, 27, 13, 14].

**Contributions:** (i) We formulate the problem of *learning specifications from demonstrations* in terms of Maximum a Posteriori inference. (ii) To make this inference well defined, we appeal to the principle of maximum entropy culminating in the distribution given (9). The main contribution of this model is that it only depends on the probability the demonstrator will successfully perform task and the probability that the task is satisfied by performing actions uniformly at random. Because these properties can be estimated without explicitly unrolling the dynamics in time, this model avoids many of the pitfalls characteristic to the curse of history. (iii) We provide an algorithm that exploits the piece-wise convex structure in our posterior model (9) to efficiently perform Maximum a Posteriori inference for the most probable specification.

**Outline:** In Sec 2, we define specifications and probabilistic automata (Markov Decision Processes without rewards). In Sec 3, we introduce the problem of *specification inference from demonstrations*, and inspired by Maximum Entropy IRL [35], develop a model of the posterior probability of a specification given a sequence of demonstrations. In Sec 4, we develop an algorithm to perform inference under (9). Finally, in Sec 5, we demonstrate how due to their inherent composability, learning specifications can avoid common bugs that often occur due to ad-hoc reward composition.

## 2   Background

We seek to learn specifications from demonstrations provided by a teacher who executes a sequence of actions that probabilistically changes the system state. For simplicity, we assume that the set of actions and states are finite and fully observed. The system is naturally modeled as a probabilistic automaton[1] formally defined below:

> **Definition 1** (Probabilistic Automaton). *A **probabilistic automaton** is a tuple $M = (S, s_0, A, \delta)$, where $S$ is the finite set of states, $s_0 \in S$ is the initial state, $A$ is the finite set of actions, and $\delta : S \times A \times S \to [0, 1]$ specifies the transition probability of going from $s$ to $s'$ given action $a$, i.e. $\delta(s, a, s') = \Pr(s' \mid s, a)$ and $\sum_{s' \in S} \Pr(s' \mid s, a) = 1$ for all states $s$.*

> **Definition 2** (Trace). *A sequence of state/action pairs is called a **trace** (trajectory, demonstration). A trace of length $\tau \in \mathbb{N}$ is an element of $(S \times A)^\tau$.*

Next, we develop machinery to distinguish between desirable and undesirable traces. For simplicity, we focus on finite trace properties, referred to as specifications, that are decidable within some fixed $\tau \in \mathbb{N}$ time steps, e.g., "event A occurred in the last 20 steps".

> **Definition 3** (Specification). *Given a set of states $S$, a set of actions $A$, and a fixed trace length $\tau \in \mathbb{N}$, a **specification** is a subset of traces $\varphi \subseteq (S \times A)^\tau$. We define $true \overset{\text{def}}{=} (S \times A)^\tau$, $\neg\varphi \overset{\text{def}}{=} true \setminus \varphi$, and $false \overset{\text{def}}{=} \neg true$. A collection of specifications, $\Phi$, is called a **concept class**. Finally, we abuse notation and use $\varphi$ to also denote its indicator function (interpreted as a non-Markovian Boolean reward),*
>
> $$\varphi(\xi) \overset{\text{def}}{=} \begin{cases} 1 & if\ \xi \in \varphi \\ 0 & otherwise \end{cases}. \tag{1}$$

Specifications may be given in formal notation, as sets or automata. Further, each representation facilitates defining a plethora of composition rules. For example, consider two specifications, $\varphi_A$, $\varphi_B$ that encode tasks $A$ and $B$ respectively and the composition rule $\varphi_A \cap \varphi_B : \xi \mapsto \min\big(\varphi_A(\xi), \varphi_B(\xi)\big)$. Because the agent only receives a non-zero reward if $\varphi_A(\xi) = \varphi_B(\xi) = 1$, a reward maximizing agent must necessarily perform tasks $A$ and $B$ simultaneously. Thus, $\varphi_A \cap \varphi_B$ corresponds to conjunction (logical *and*). Similarly, maximizing $\varphi_A \cup \varphi_B : \xi \mapsto \max\big(\varphi_A(\xi), \varphi_B(\xi)\big)$ corresponds to disjunction (logical *or*). One can also encode conditional requirements using subset inclusion, e.g., maximizing $\varphi_A \subseteq \varphi_B : \xi \mapsto \max\big(1 - \varphi_A(\xi), \varphi_B(\xi)\big)$ corresponds to task A triggering task B.

Complicated temporal connectives can also be defined using temporal logics [25] or automata [32]. For our purposes, it suffices to informally extend propositional logic with three temporal operators: (1) Let $Ha$, read "historically $a$", denote that property $a$ held at all previous time steps. (2) Let $Pa \overset{\text{def}}{=} \neg(H\neg a)$, read "once $a$", denote that the property $a$ at least once held in the past. (3) Let $a\ S\ b$, read "$a$ since $b$", denote that the property $a$ that has held every time step after $b$ last held. The true power of temporal operators is realized when they are composed to make more complicated sentences. For example, $H(a \implies (b\ S\ c))$ translates to "it was always the case that if $a$ was true, then $b$ has held since the last time $c$ held.". Observe that the property BBY from the introductory example takes this form, $H((\text{yellow} \wedge P\ \text{blue}) \implies (\neg\text{blue}\ S\ \text{brown}))$, i.e., "Historically, if the agent had once visited blue and is currently visiting yellow, then the agent has not visited blue since it last visited brown".

# 3 Specification Inference from Demonstrations

In the spirit of Inverse Reinforcement Learning, we now seek to find the specification that best explains the behavior of the agent. We refer to this as *Specification Inference from Demonstrations*.

> **Definition 4** (Specification Inference from Demonstrations). *The **specification inference from demonstrations** problem is a tuple $(M, X, \Phi)$ where $M = (S, s_0, A, \delta)$ is a probabilistic automaton, $X$ is a (multi-)set of $\tau$-length traces drawn from an unknown distribution induced by a teacher attempting to demonstrate some unknown specification within $M$, and $\Phi$ a concept class of specifications.*
>
> *A solution to $(M, X, \Phi)$ is:*
> $$\varphi^* \in \underset{\varphi \in \Phi}{\arg\max} \Pr(\varphi \mid M, X) \tag{2}$$
>
> *where $\Pr(\varphi \mid M, X)$ denotes the probability that the teacher demonstrated $\varphi$ given the observed traces, $X$, and the dynamics, $M$.*

To make this inference well-defined, we make a series of assumptions culminating in (9).

**Likelihood of a demonstration:** We begin by leveraging the principle of maximum entropy to disambiguate the likelihood distributions. Concretely, define:

$$w\big(\xi = (\mathbf{s}, \mathbf{a}), M\big) = \prod_{i=0}^{\tau-1} \Pr(s_{i+1} | s_i, a_i, M) \tag{3}$$

where $\mathbf{s}$ and $\mathbf{a}$ are the projected sequences of states and actions of $\xi$ respectively, to be the weight of each possible demonstration $\xi$ induced by dynamics $M$. Given a demonstrator who on average satisfies the specification $\varphi$ with probability $\overline{\varphi}$, we approximate the likelihood function by:

$$\Pr\left(\xi \mid M, \varphi, \overline{\varphi}\right) = w(\xi, M) \cdot \frac{\exp(\lambda_\varphi \varphi(\xi))}{Z_\varphi} \tag{4}$$

where $\lambda_\varphi$, $Z_\varphi$ are normalization factors such that $\mathbb{E}_\xi[\varphi] = \overline{\varphi}$ and $\sum_\xi \Pr(\xi \mid M, \varphi) = 1$. For a detailed derivation that (4) is the maximal entropy distribution, we point the reader to [18]. Next observe that due to the Boolean nature of $\varphi$, (4) admits a simple closed form:

$$\boxed{\Pr(\xi \mid M, \varphi, \overline{\varphi}) = \widetilde{\{\xi\}} \cdot \begin{cases} \overline{\varphi}/\widetilde{\varphi} & \xi \in \varphi \\ (1 - \overline{\varphi})/\widetilde{\neg\varphi} & \xi \notin \varphi \end{cases}} \tag{5}$$

where in general we use $\widetilde{(\cdot)}$ to denote the probability of satisfying a specification using uniformly random actions. Thus, we denote by $\widetilde{\{\xi\}}$ the probability of randomly generating demonstration $\xi$ within $M$. Further, note that by the law of the excluded middle, for any specification: $\widetilde{\neg\varphi} = 1 - \widetilde{\varphi}$.

*Proof Sketch.* For brevity, let $W_\varphi \overset{\text{def}}{=} \sum_{\xi \in \varphi} w(\xi, M)$ and $c \overset{\text{def}}{=} e^{\lambda_\varphi}$. Via the constraints on (4),

$$Z_\varphi \cdot \overline{\varphi} = 1 \cdot \sum_{\xi \in \varphi} c^1 \cdot w(\xi, M) + 0 \cdot \sum_{\xi \notin \varphi} c^0 \cdot w(\xi, M) = cW_\varphi$$
$$Z_\varphi = c^1 \sum_{\xi \in \varphi} w(\xi, M) + c^0 \sum_{\xi \notin \varphi} w(\xi, M) = cW_\varphi + W_{\neg\varphi} \tag{6}$$

Combining gives $Z_\varphi = W_{\neg\varphi}/(1 - \overline{\varphi})$. Next, observe that if $\xi \notin \varphi$, then $e^{\lambda_\varphi \varphi(\xi)} = 1$ and substituting in (4) yields, $\Pr(\xi \mid \varphi, M, \xi \notin \varphi) = w_\xi(1 - \overline{\varphi})/W_{\neg\varphi}$. If $\xi \in \varphi$ (implying $W_\varphi \neq 0$) then $e^{\lambda_\varphi} = Z_\varphi \overline{\varphi}/W_\varphi$ and $\Pr(\xi \mid \varphi, M, \xi \in \varphi) = w_\xi \overline{\varphi}/W_\varphi$. Finally, observe that $\widetilde{\varphi} = W_\varphi/W_{\text{true}}$ and $\widetilde{\{\xi\}} = w_\xi/W_{\text{true}}$. Substituting and factoring yields (5). $\qquad\square$

**Likelihood of a set of demonstrations:** If the teacher gives a finite sequence of $\tau$ length demonstrations, $X$, drawn i.i.d. from (5), then the log likelihood, $\mathcal{L}$, of $X$ under (5) is:[2]

$$\mathcal{L}(X \mid M, \varphi, \overline{\varphi}) = \log\left(\prod_{\xi \in X} \widetilde{\{\xi\}}\right) + N_\varphi \ln\left(\frac{\overline{\varphi}}{\widetilde{\varphi}}\right) + N_{\neg\varphi} \ln\left(\frac{\overline{\neg\varphi}}{\widetilde{\neg\varphi}}\right) \tag{7}$$

where by definition we take $(0 \cdot \ln(\ldots) = 0)$ and $N_\varphi \overset{\text{def}}{=} \sum_{\xi \in X} \varphi(\xi)$.

Next, observe that $\left[\overline{\varphi}\ln\left(\frac{\overline{\varphi}}{\widetilde{\varphi}}\right) + (1 - \overline{\varphi})\ln\left(\frac{1 - \overline{\varphi}}{1 - \widetilde{\varphi}}\right)\right]$ is the information gain (KL divergence) between two Bernoulli distributions with means $\overline{\varphi}$ and $\widetilde{\varphi}$ resp. Syntactically, let $\mathcal{B}(\mu)$ denote a Bernoulli distribution with mean $\mu$ and $D_{KL}(P \parallel Q) \overset{\text{def}}{=} \sum_i P(i) \ln(P(i)/Q(i))$ denote the information gain when using distribution $P$ compared to $Q$. If $X$ is "representative" such that $N_\varphi \approx \overline{\varphi} \cdot |X|$, we can (up to a $\varphi$ independent normalization) approximate (7):

$$\Pr(X \mid M, \varphi, \overline{\varphi}) \overset{\propto}{\sim} \exp\left(|X| \cdot D_{KL}\left(\mathcal{B}(\overline{\varphi}) \parallel \mathcal{B}(\widetilde{\varphi})\right)\right) \tag{8}$$

Where $\overset{\propto}{\sim}$ denotes approximately proportional to. Unfortunately, the approximation $|X| \cdot \overline{\varphi} \approx N_\phi$ implies that, $\overline{\neg\varphi} = 1 - \overline{\varphi}$ which introduces the undesirable symmetry, $\Pr(X \mid M, \varphi, \overline{\varphi}) = \Pr(X \mid M, \neg\varphi, \overline{\neg\varphi})$, into (8). To break this symmetry, we assert that the demonstrator must be at least as good as random. Operationally, we assert that $\Pr(\varphi \mid \overline{\varphi} < \widetilde{\varphi}) = 0$ and is otherwise uniform. Finally, we arrive at the posterior distribution given in (9), where $\mathbf{1}[\cdot]$ denotes an indicator function.

$$\boxed{\Pr(\varphi \mid M, X, \overline{\varphi}) \overset{\propto}{\sim} \overbrace{\mathbf{1}[\overline{\varphi} \geq \widetilde{\varphi}]}^{\text{Demonstrator is better than random.}} \cdot \exp\left(|X| \cdot \overbrace{D_{KL}\left(\mathcal{B}(\overline{\varphi}) \parallel \mathcal{B}(\widetilde{\varphi})\right)}^{\text{Information gain over random actions.}}\right)} \tag{9}$$

[2]We have suppressed a multinomial coefficient required if any two demonstrations are the same. However, this term will not change as $\varphi$ varies, and thus cancels when comparing across specifications.

# 4    Algorithm

In this section, we exploit the structure imposed by (9) to efficiently search for the most probable specification (2) within a (potentially large) concept class, $\Phi$. Namely, observe that under (9), the specification inference problem (2) reduces to maximizing the information gain over random actions.

$$\varphi^* \in \underset{\varphi \in \Phi}{\arg\max} \left\{ \mathbf{1}[\overline{\varphi} \geq \widetilde{\varphi}] \cdot D_{KL}\Big( \mathcal{B}(\overline{\varphi}) \parallel \mathcal{B}(\widetilde{\varphi}) \Big) \right\} \tag{10}$$

Because gradients on $\widetilde{\varphi}$ and $\overline{\varphi}$ are not well-defined, gradient descent based algorithms are not applicable. Further, while evaluating if a trace satisfies a specification is fairly efficient (and thus our $N_\varphi/|X|$ approximation to $\overline{\varphi}$ is assumed easy to compute), computing $\widetilde{\varphi}$ is in general known to be $\#P$-complete [2]. Nevertheless, in practice, moderately efficient methods for computing or approximating $\widetilde{\varphi}$ exist including Monte Carlo simulation [22] and weighted model counting [5] via Binary Decision Diagrams (BDDs) [3] or repeated SAT queries [4]. As such, we seek an algorithm that poses few $\widetilde{\varphi}$ queries. We begin with the observation that adding a trace to a specification cannot lower its probability of satisfaction under random actions.

> **Lemma 1.** $\forall \varphi', \varphi \in \Phi \, . \, \varphi' \subseteq \varphi$ implies $\widetilde{\varphi}' \leq \widetilde{\varphi}$ and $\overline{\varphi}' \leq \overline{\varphi}$.
>
> *Proof.* The probability of sampling an element of a set monotonically increases as elements are added to the set independent of the *fixed* underlying distribution over elements.  □

Further, note that $N_\varphi$ (and thus, our approximation to $\overline{\varphi}$) can only take on $|X| + 1$ possible values. This suggests a piece-wise analysis of (10) by conditioning on the value of $\overline{\varphi}$.

> **Definition 5.** *Given candidate specifications $\Phi$ and a subset of demonstrations $S \subseteq X$ define,*
>
> $$\Phi_S \overset{\text{def}}{=} \{\varphi \in \Phi \, : \, \varphi \cap X = S\} \qquad J_{|S|}(x) \overset{\text{def}}{=} \mathbf{1}\left[\frac{|S|}{|X|} \geq x\right] \cdot D_{KL}\left(\mathcal{B}(\frac{|S|}{|X|}) \parallel \mathcal{B}(x)\right) \tag{11}$$

The next key observation is that $J_{|S|} : [0, 1] \to \mathbb{R}_{\geq 0}$ monotonically decreases in $x$.

> **Lemma 2.** $\forall S \subseteq X, \, x < x' \implies J_{|S|}(x) \leq J_{|S|}(x')$
>
> *Proof.* To begin, observe that $D_{KL}$ is always non-negative. Due to the $\mathbf{1}[\frac{|S|}{|X|} \geq x]$ indicator, $J_{|S|}(x) = 0$ for all $x > |S|/|X|$. Next, observe that $J_{|S|}$ is convex due to the convexity of the $D_{KL}$ on Bernoulli distributions and is minimized at $x = |S|/|X|$ (KL Divergence of identical distributions is 0). Thus, $J_{|S|}(x)$ monotonically decreases as $x$ increases.  □

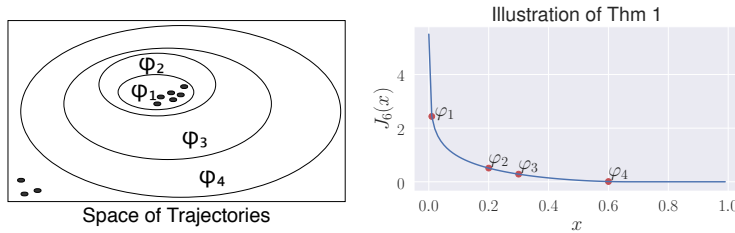

Figure 3: Left: An example of a series of specifications $\varphi_1, \ldots, \varphi_4$ ordered by subset inclusion. The dots represent demonstrations, and so each specification has $\overline{\varphi}_i = 6/9$. Right: Plot of $J_{|S|}(x)$ for hypothetical values of $\widetilde{\varphi}_i$ annotated as points. Notice that the sequence of specifications is ordered on the $x$-axis, and thus the maximum must occur at the start of the sequence.

These insights are then combined in Theorem 1 and illustrated in Fig 3.

> **Theorem 1.** If $A$ denotes a sequence of specifications, $\varphi_1, \ldots, \varphi_n$, ordered by subset inclusion $j \leq k \implies \varphi_j \subseteq \varphi_k$ and $S \subseteq X$ is an arbitrary subset of demonstrations, then:
>
> $$\max_{\varphi \in A} J_{|S|}(\widetilde{\varphi}) = J_{|S|}(\widetilde{\varphi}_1) \tag{12}$$

*Proof.* $\widetilde{\varphi}$ is monotonically increasing on $A$ (Lemma 1). Via Lemma 2 $J_{|S|}(x)$ is monotonically decreasing and thus the maximum of $J_{|S|}(\widetilde{\varphi})$ must occur at the beginning of $A$. $\square$

**Lattice Concept Classes.**

Theorem 1 suggests specializing to concept classes where determining subset relations is easy. We propose studying concept classes organized into a finite (bounded) lattice, $(\Phi, \trianglelefteq)$, that respects subset inclusion: $(\varphi \trianglelefteq \varphi' \implies \varphi \subseteq \varphi')$. To enforce the bounded constraint, we assert that *true* and *false* are always assumed to be in $\Phi$ and act as the bottom and top of the partial order respectively. Intuitively, this lattice structure encodes our partial knowledge of which specifications imply other specifications. These implication relations can be represented as a directed graph where the nodes correspond to elements of $\Phi$ and an edge is present if the source is known to imply the target. Because implication is transitive, many of the edges can be omitted without losing any information. The graph resulting from this transitive reduction is called a Hasse diagram [6] (See Fig 4). In terms of the graphical model, the Hasse diagram encodes that for certain pairs of specifications, $\varphi, \varphi'$, we know that $\Pr(\varphi(\xi) = 1 \mid \varphi'(\xi) = 1, M) = 1$ or $\Pr(\varphi(\xi) = 0 \mid \varphi'(\xi) = 0, M) = 1$.

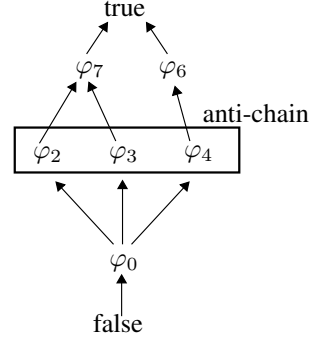

Figure 4: Hasse Diagram of an example lattice $\Phi$ with an anti-chain annotated. Directed edges represent known subset relations and paths represent chains.

**Inference on chain concept classes.** Sequences of specifications ordered by subset inclusion generalize naturally to ascending chains.

---

**Definition 1** (Chains and Anti-Chains)**.** Given a partial order $(\Phi, \trianglelefteq)$, an ascending *chain* (or just chain) is a sequence of elements of $A$ ordered by $\trianglelefteq$. The smallest element of the chain is denoted, $\downarrow(A)$. Finally, an *anti-chain* is a set of incomparable elements. An anti-chain is called maximal if no element can be added to it without making two of its elements comparable.

---

Recasting Theorem 1 in the parlance of chains yields:

---

**Corollary 1.** If $S \subseteq X$ is a subset of demonstrations and $A$ is a chain in $(\Phi_S, \trianglelefteq)$ then:

$$\max_{\varphi \in A} J_{|S|}(\widetilde{\varphi}) = J_{|S|}(\widetilde{\downarrow(A)}) \tag{13}$$

---

Observe that if the lattice, $(\Phi, \trianglelefteq)$ is itself a chain, then there are at most $|X| + 1$ non-empty demonstration partitions, $\Phi_S$. In fact, the non-empty partitions can be re-indexed by the cardinality of $S$, e.g., $\Phi_S \mapsto \Phi_{|S|}$. Further, note that since chains are totally ordered, the smallest element of each non-empty partition can be found by performing a binary search (indicated by find_smallest below). These insights are combined into Algorithm 1 with a relativized run-time analysis given in Thm 2.

---

**Algorithm 1** Inference on chains

1: **procedure** CHAIN_INFERENCE($X, (A, \trianglelefteq)$)
2:    $\Psi \leftarrow \left\{ (i, \text{find\_smallest}(A, i)) \mid i \in \{0, 1, \ldots, |X|\} \right\}$    $\triangleright O(|T_{\text{data}}||X|\ln(|A|))$.
3:    **return** $i, \varphi^* \leftarrow \underset{i, \varphi \in \Psi}{\arg\max} J_i(\widetilde{\varphi})$    $\triangleright O(T_{\text{rand}}|X|)$

---

**Theorem 2.** Let $T_{\text{data}}$ and $T_{\text{rand}}$ respectively represent the worst case execution time of computing $\overline{\varphi}$ and $\widetilde{\varphi}$ for $\varphi$ in chain $A$. Given demonstrations $X$, Alg 1 runs in time:

$$O\left( |X| \left( T_{\text{data}} \ln(|A|) + T_{\text{rand}} \right) \right) \tag{14}$$

---

*Proof Sketch.* A binary search over $|A|$ elements takes $\ln(|A|)$ time. There are $|X|$ binary searches required to find the smallest element of each partition. Finally, for each smallest element, a single random satisfaction query is made. ∎

**Lattice inference.** Of course, in general, $(\Phi, \trianglelefteq)$ is not a chain, but a complicated lattice. Nevertheless, observe that any path from *false* to *true* is a chain. Further, the smallest element of each partition must either be the same specification or incomparable in $(\Phi, \trianglelefteq)$. That is, for each $k \in \{0, 1, \dots |X|\}$, the set:

$$B_k \overset{\text{def}}{=} \left\{ \downarrow (\Phi_S) \ : \ S \in \binom{X}{k} \right\} \tag{15}$$

is a maximal anti-chain. Thus, Corollary 1 can be extended to:

**Corollary 2.** Given a lattice $(\Phi, \trianglelefteq)$ and demonstrations $X$:

$$\max_{\varphi \in \Phi} J_{N_\varphi}(\widetilde{\varphi}) = \max_{k \in 0,1,\dots,|X|} \max_{\varphi \in B_k} J_k(\widetilde{\varphi}) \tag{16}$$

Recalling that $N_\varphi$ increases on paths from *false* to *true*, we arrive at the following simple algorithm which takes as input the demonstrations and the lattice $\varphi$ encoded as a directed acyclic graph rooted at *false*. (i) Perform a breadth first traversal (BFT) of the lattice $(\Phi, \trianglelefteq)$ starting at *false* (ii) During the traversal, if specification $\varphi$ has a larger $N_\varphi$ than all of its direct predecessors, then check if it is more probable than the best specification seen so far (if so, make it the most probable specification seen so far). (iii) At the end of the traversal, return the most probable specification. Pseudo code is provided in Algorithm 2 with a run-time analysis given in Theorem 3.

---

**Algorithm 2** Inference on Partial Orders

---

1: **procedure** PARTIALORDER_INFERENCE$(X, (\Phi, \trianglelefteq))$
2:    $(\varphi^*, \text{best\_info\_gain}) \leftarrow (false, 0)$
3:    **for** $\varphi$ in breadth_first_traversal$((\Phi, \trianglelefteq))$ **do**
4:        parents $\leftarrow$ direct_predecessors$(\varphi)$
5:        **if** $\exists \varphi' \in$ parents $.\ N_{\varphi'} = N_\varphi$ **then**
6:            **continue**
7:        info_gain $\leftarrow J_{N_\varphi}(\widetilde{\varphi})$
8:        **if** info_gain $>$ best_info_gain **then**
9:            $(\varphi^*, \text{best\_info\_gain}) \leftarrow (\varphi, \text{info\_gain})$
10:    **return** $\varphi^*$

---

**Theorem 3.** Let $(\Phi, \trianglelefteq)$ be a bounded partial order encoded as a Directed Acyclic Graph (DAG), $G = (V, E)$, with vertices $V$ and edges $E$. Further, let $B$ denote the largest anti-chain in $\Phi$. If $T_{\text{data}}$ and $T_{\text{rand}}$ respectively represent the worst case execution time of computing $\overline{\varphi}$ and $\widetilde{\varphi}$, then for demonstrations $X$, Alg 2 runs in time:

$$O\big(E + T_{\text{data}} \cdot V + T_{\text{rand}} \cdot |B||X|\big) \tag{17}$$

*Proof sketch.* BFT takes $O(V + E)$. Further, for each node, $\overline{\varphi}$ is computed $(O(T_{\text{data}} \cdot V))$. Finally, for each node in each of the candidate anti-chains $B_k$, $\widetilde{\varphi}$ is computed. Since $|B|$ is the size of the largest anti-chain, this query happens no more than $|B||X|$ times. ∎

## 5  Experiments and Discussion

**Scenario.** Recall our introductory gridworld example Ex 1. Now imagine that the robot is pre-programmed to perform task the "recharge and avoid lava" task, but is unaware of the second requirement, "do not recharge when wet". To signal this additional constraint to the robot, the human operator provides the five demonstrations shown in Fig 1. We now illustrate how learning specifications rather than Markovian rewards enables the robot to safely compose the new constraint with its existing knowledge to perform the joint task in a manner that is robust to changes in the task.

To begin, we assume the robot has access to the Boolean features: red (lava tile), blue (water tile), brown (drying tile), and yellow (recharge tile). Using these features, the robot has encoded the "recharge and avoid lava" task as: $H(\neg \text{red}) \wedge P(\text{yellow})$.

**Concept Class.** We designed the robot's concept class to be the conjunction of the known requirements and a specification generated by the grammar on the right. The motivation in choosing this grammar was that (i) it generates a moderately large concept class (930 possible specifications after pruning trivially false specifications), and (ii) it contains several interesting alternative tasks such as $H(\text{red} \implies (\neg\text{brown } S \text{ blue}))$, which semantically translates to: "the robot should be wet before entering lava". To generate

> *Concept Class Grammar:*
>
> $\langle\phi\rangle \models \langle\text{H } \psi\rangle \mid \langle\text{P } \psi\rangle$
> $\langle\psi\rangle \models \langle\beta\rangle \mid \langle\beta\rangle \implies \langle\beta\rangle$
> $\langle\beta\rangle \models \langle\alpha\rangle \mid \langle\alpha\rangle \wedge \langle\alpha\rangle \mid \langle\alpha\rangle \text{ S } \langle\alpha\rangle$
> $\langle\alpha\rangle \models AP \mid \neg AP$
> $\langle AP\rangle \models yellow \mid red \mid brown \mid blue$

the edges in Hasse diagram, we unrolled the formula into their corresponding Boolean formula and used a SAT solver to determine subset relations. While potentially slow, we make three observations regarding this process: (i) the process was trivially parallelizable (ii) so long as the atomic predicates remain the same, this Hasse diagram need not be recomputed since it is otherwise invariant to the dynamics (iii) most of the edges in the resulting diagram could have been syntactically identified using well known identities on temporal logic formula.

**Computing $\widetilde{\varphi}$.** To perform random satisfaction rate queries, $\widetilde{\varphi}$, we first ran Monte Carlo to get a coarse estimate and we symbolically encoded the dynamics, color sensor, and specification into a Binary Decision Diagram to get exact values. This data structure serves as an incredibly succinct encoding of the specification aware unrolling of the dynamics, which in practice avoids the exponential blow up suggested by the curse of history. We then counted the number of satisfying assignments and divided by the total possible number of satisfying assignments.[3] On average in these candidate pools, each query took $0.4$ seconds with a standard deviation of $0.32$ seconds.

**Results.** Running a fairly unoptimized implementation of Algorithm 2 on the concept class and demonstrations took approximately 95 seconds and resulted in 172 $\widetilde{\varphi}$ queries ($\approx 18\%$ of the concept class). The inferred additional requirement was $H((\text{yellow} \wedge P \text{ blue}) \implies (\neg\text{blue } S \text{ brown}))$ which exactly captures the do not recharge while wet constraint. Compared to a brute force search over the concept class, our algorithm offered an approximately 5.5 fold improvement. Crucially, there exists controllable trajectories satisfying the joint specification:

$$\left(H\neg\text{red} \wedge P \text{ yellow}\right) \wedge H\left((\text{yellow} \wedge P \text{ blue}) \implies (\neg\text{blue } S \text{ brown})\right). \tag{18}$$

Thus, a specification optimizing agent *must* jointly perform both tasks. This holds true even under task changes such as that in Fig 2. Further, observe that it was fairly painless to incorporate the previously known recharge while avoiding lava constraints. Thus, in contrast to quantitative Markovian rewards, learning Boolean specifications enabled encoding *compositional temporal specifications* that are *robust* to changes in the environment.

# 6 Conclusion and Future work

Motivated by the problem of compositionally learning from demonstrations, we developed a technique for learning binary non-Markovian rewards, which we referred to as *specifications*. Because of their limited structure, specifications enabled first learning sub-specifications for subtasks and then later creating a composite specifications that encodes the larger task. To learn these specifications from demonstrations, we applied the principle of maximum entropy to derive a novel model for the likelihood of a specification given the demonstrations. We then developed an algorithm to efficiently search for the most probable specification in a candidate pool of specifications in which some subset relations between specifications are known. Finally, in our experiment, we gave a concrete instance where using traditional learning composite reward functions is non-obvious and error-prone, but inferring specifications enables trivial composition. Future work includes extending the formalism to infinite horizon specifications, continuous dynamics, characterizing the optimal set of teacher demonstrations under our posterior model [34], efficiently marginalizing over the whole concept class and exploring alternative data driven methods for generating concept classes.

**Acknowledgments.** We would like to thank the anonymous referees as well as Daniel Fremont, Markus Rabe, Ben Caulfield, Marissa Ramirez Zweiger, Shromona Ghosh, Gil Lederman, Tommaso Dreossi, Anca Dragan, and

Natarajan Shankar for their useful suggestions and feedback. The work of the authors on this paper was funded in part by the US National Science Foundation (NSF) under award numbers CNS-1750009, CNS-1740079, CNS-1545126 (VeHICaL), the DARPA BRASS program under agreement number FA8750–16–C0043, the DARPA Assured Autonomy program, by Toyota under the iCyPhy center and the US ARL Cooperative Agreement W911NF-17-2-0196.

## Footnotes

[1]Probabilistic Automata are often constructed as a Markov Decision Process, $M$, without its Markovian reward map $R$, denoted $M \setminus R$.

[3]One can add probabilities to transitions by adding to transition constraints additional fresh variables such that the number of satisfying assignments is proportional to the probability.

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
