[Reviews · NeurIPS 2018]

Reviewer 1



** update ** I increase the score I attribute to this paper since the authors' response to the various problems raised by the reviewers seems to me very satisfactory. However the authors promise several improvements and my score cannot be based solely on promises, so this score only increases by one. Finally, the practical motivations of this little explored field are extremely justified, so I also encourage authors to apply their work to more complex problems, which should lead to interesting results. ** summary ** This paper sets up an algorithm capable from demonstrations (sequence of states and actions) and from a set of specifications (set of trajectories), to infer the specifications governing the actions of the agent and the visited states. These specifications can be seen as non-Markovian reward functions. Thus, this work is related to inverse reinforcement learning (IRL) which aims to infer the reward function of an agent by observing these successive states and actions. By defining the probability of a trajectory knowing a specification (using the maximum entropy principle) the development leads to a posterior distribution. Two algorithms result from this and allow to test the approach on the system presented in introduction (motivating the paper). ** quality ** Although its introduction is interesting, this paper is quite hard to follow, especially from page 4. The steps should be better justified or motivated by intuition or pedagogical explanations of the ongoing development. Also some proofs only appears in the supplementary materials and not in the paper, making the paper less self-contained. At least sketches of proof should be provided to let the reader study the main arguments. ** clarity ** As mentioned above, clarity is not the strength of this paper. This can also be illustrated by the more or less in-depth comments that follow. page 3: < Sec > Section The Markov property is already present in Definition 2.1. < no \phi =^{def} 1 - \phi(\xi) > no \phi(\xi) =^{def} 1 - \phi(\xi) page 4: In Equation 2, it seems that we multiply a probability of \xi (trajectory probability given by the transition functions) with another probability of \xi (given by the exponential function). Some confusion in the notations X and \xi. page 6: < the following simple algorithm. > the following simple algorithm (Algorithm 2). page 8: < queries,\phi^{hat} > queries, \phi^{hat} < assignments.\footnote{...} > assignments\footnote{...}. The conclusion should be a Section. ** originality ** To the best of my knowledge this paper is the first research on inferring specifications from demonstrations in the MDP framework. The authors have done a good bibliographical work to position their study in the relevant literature. I suggest authors read (Drougard "Exploiting Imprecise Information Sources in Sequential Decision Making Problems under Uncertainty." 2015) which adapts classical MDP models to Qualitative Possibility Theory. This theory is very close to logic (and therefore could easily integrate the specifications framework) while retaining properties similar to Probability Theory (which allows planning with the same tools). ** significance ** Although the idea and the algorithms developed are really interesting, I doubt that the theoretical and applied contributions are worthy of NIPS. I remain cautious however because the lack of clarity, the absence of sketch of proof and the short time to review prevents me from checking developments in details.

Reviewer 2



This paper proposes a method for learning specifications (binary non-Markovian rewards defined as predicates over trajectories) from demonstrations. The motivation is to address some problems with reward function design, specifically to provide a better way combine rewards for different subtasks. They formally specify the problem of inferring specifications from demonstrations, define an algorithm inspired by Maximum Entropy IRL, and present theoretical results and experiments. The paper is interesting, clearly written and relatively well-motivated. Overall, I think the method has strong theoretical grounding but weak experimental support. The experiment section does not contain all the experiments and details needed to back up the claims of the paper (and neither does the supplementary material). They test MaxEnt IRL and the proposed specification inference algorithm on a gridworld with non-Markovian dependencies, where the agent needs to enter a drying tile after stepping into water before going to the goal. Figure 1 shows rewards inferred by MaxEnt IRL for the two subtasks and the whole task, and the text describes the behavior of agents optimizing these subtask rewards or various combinations of these rewards, but does not mention how the agent optimizing the reward inferred for the whole task (Figure 1b) would behave. The paper claims that the rewards learned using IRL would lead to unsafe behavior, but does not include the corresponding experimental results. The paper shows results for specifications learned by the proposed algorithm, but not for the agent behavior that results from optimizing for these specifications. No details are given on the type of agent used in these experiments. The paper states that "we demonstrate how learning specifications can help avoid common reward hacking bugs", but there are no experimental results for an agent using the learned specifications to support this claim. I also think the authors are using a nonstandard definition of "reward hacking", which includes any agent behavior that results from optimizing a poorly designed reward function, while the standard definition (given in Amodei et al, 2016) refers to the agent finding a loophole in an otherwise well designed reward function. It would be good to see experimental results on environments designed to test for reward hacking, such as the "boat race" environment from the AI Safety Gridworlds suite (Leike et al, 2017). UPDATE after reading the author feedback: - I'm happy to see that most of the missing details for the experiment have been provided. - The argument for why this method would help avoid reward hacking in the boat race scenario seems convincing. I would still like to see experimental results on this environment, which should be easy to obtain. - Overall, the rebuttal has addressed most of my concerns, so I have increased my score.

Reviewer 3



This paper addressed the limitation of scalar reward functions in inverse reinforcement learning and attempted to capture the temporal dependency and compositionality in task specification. The proposed approach uses temporal logic equations to represent binary non-Markovian rewards, where compose specifications can be formed via logic operators. The draft is well written. Though the technical description is quite dense, the authors did a good job laying out the background knowledge and introducing the algorithms and theorems in a clear and organized way. To infer the specifications from demonstrations, they built a posterior model, so that finding the best specification is cast as a MAP inference problem. However, as the gradients are not well defined for some components of the model, they developed a search-based method to find the most probable specification from a candidate pool. This search-based method required manual generation of a candidate pool of specifications in order to create a Hasse diagram of all specifications, where search is performed. The method only scores the specifications from the candidate pool. So its performance heavily relies on how the manual generation is done. I wonder if data-driven methods can be exploited to automatically generate such a candidate pool of specifications. In addition, the authors have only demonstrated the proposed approach in a toy-sized problem described in the introduction. It is unclear of its extensibility and scalability in larger problem instances and more diverse domains. In the toy task used in this paper, the authors created two Hasse diagrams, each with 127 nodes. In this case, the search method has a threefold speedup over the brute force approach. The size of candidate pools and the Hasse diagrams would be significantly larger to tackle more realistic and complex tasks, leading to a large increase of computation time. The binary Boolean specifications complement the scalar reward functions in its flexibility to capture the time dependency and compositional task structure. However, it might not have the expressive power to represent all task objectives, e.g., running forward as fast as possible. I wonder if the Boolean specifications can be combined with the quantitative reward functions to get the best of both worlds. The toy example demonstrated the way of finding the optimal specifications in a simplistic grid-world domain. But it is unclear how the logic specifications can be defined in tasks where the subtask boundaries are indefinite, such as continuous control. Given the concerns regarding the extensibility and scalability of the proposed approach, this work could bear a greater significance with some showcase of more diverse and large-scale domains. Minor: Typo: Line 13: “a specifications” --> “specifications”